
# Comparison of temperature dependent calibration methods of an instrument to measure OH and HO₂ radicals using laser-induced fluorescence spectroscopy

**Frank A. F. Winiberg[1,2], William J. Warman[1], Charlotte A. Brumby[1], Graham Boustead[1], Iustinian G. Bejan[1,3], Thomas H. Speak[1], Dwayne E. Heard[1], Daniel Stone[1] and Paul W. Seakins[1]**

[1] School of Chemistry, University of Leeds, Leeds, LS2 9JT, United Kingdom

[2] Now at: NASA's Jet Propulsion Laboratory, California Institute of Technology, Pasadena, 91109, USA

[3] Now at: Faculty of Chemistry and "Integrated Centre for Environmental Science Studies in the North-East Development Region – CERNESIM", "Al. I. Cuza" University of Iasi, Romania

*Correspondence to*: Paul W. Seakins (p.w.seakins@leeds.ac.uk)

**Abstract**

Laser Induced Fluorescence (LIF) spectroscopy has been widely applied to fieldwork measurements of OH radicals, and of HO₂, following conversion to OH, over a wide variety of conditions, on different platforms, and in simulation chambers. Conventional calibration of HOₓ (OH + HO₂) instruments has mainly relied on a single method, generating known concentrations of HOₓ from H₂O vapour photolysis in a flow of zero air impinging just outside the sample inlet ($S_{HOx} = C_{HOx}.[HOx]$, where $S_{HOx}$ is the observed signal and $C_{HOx}$ is the calibration factor). The FAGE (Fluorescence Assay by Gaseous Expansion) apparatus designed for HOₓ measurements in the Highly Instrumented Reactor for Atmospheric Chemistry (HIRAC) at the University of Leeds has been used to examine the sensitivity of FAGE to external gas temperatures (266 – 348 K).

The conventional calibration methods give the temperature dependence of $C_{OH}$ (relative to the value at 293 K) of $(0.0059 \pm 0.0015)$ K⁻¹ and $C_{HO2}$ of $(0.014 \pm 0.013)$ K⁻¹. Errors are 2σ. $C_{OH}$ was also determined by observing the decay of hydrocarbons (typically cyclohexane) caused by OH reactions giving $C_{OH}$ (again, relative to the value at 293 K) of $(0.0038 \pm 0.0007)$ K⁻¹. Additionally, $C_{HO2}$ was determined based on the second order kinetics of HO₂ recombination with the temperature dependence of $C_{HO2}$, relative to 293 K being $(0.0064 \pm 0.0034)$ K⁻¹.



1 The temperature dependence of $C_{HOx}$ depends on HOx number density, quenching, relative

2 population of the probed OH rotational level and HOx transmission from inlet to detection axis.

3 The first three terms can be calculated and, in combination with the measured values of $C_{HOx}$,

4 show that HOx transmission increases with temperature. Comparisons with other instruments

5 and the implications of this work are discussed.

7 **1 Introduction**

8 Hydroxyl radicals (OH) play a key role in our atmosphere, oxidising a broad range of species.

9 OH is the main daytime oxidant in the troposphere and the main sink for methane, a potent

10 greenhouse gas. The OH radical is linked to the $HO_2$ radical through the oxidation of most

11 other non-methane hydrocarbons (NMHCs) and CO in the troposphere and, through reaction

12 with $NO_2$, in the upper troposphere/lower stratosphere. Due to the high reactivity of OH

13 (lifetime ~1 s even in clean air), these radicals undergo minimal transport and local

14 concentrations depend only on the *in situ* chemistry. Measurements of HOx concentrations, in

15 conjunction with measurements of their sources and sinks are a sensitive test of chemical

16 models. Accurate measurement of [HOx] is therefore paramount, not only for field

17 measurements, (Stone et al., 2012;Heard and Pilling, 2003;Gligorovski et al., 2015), but also

18 for atmospheric simulation chambers where OH/$HO_2$ instruments have been deployed (Karl et

19 al., 2004;Glowacki et al., 2007).

20 Sensitive detection techniques with high temporal resolution are required for HOx detection

21 and techniques have been reviewed in Stone et al. (2012) and Wang et al. (2021). Fluorescence

22 Assay by Gaseous Expansion (FAGE) (e.g. Hard et al. (1984)) is the most common method

23 used for both field and chamber studies. Here, the sample is expanded to low pressures and OH

24 detected by resonance fluorescence at ~308 nm. The low pressures are required to temporally

25 separate fluorescence from the excitation laser pulse. $HO_2$ is converted to OH by reaction with

26 NO and detected in a separate cell. Both techniques require calibration which is conventionally

27 based on the generation of OH and $HO_2$ from water vapour photolysis at 185 nm at atmospheric

28 temperature and pressure.

29 Recent studies have demonstrated potential interferences for measurements of both OH and

30 $HO_2$ radicals using the FAGE technique, with the magnitude dependent upon instrument design

31 (Mao et al., 2012;Novelli et al., 2014;Novelli et al., 2017;Fuchs et al., 2011;Whalley et al.,

32 2013;Fuchs et al., 2016). Considerable effort has been made to minimize, understand and



mitigate any interference, with many groups now fitting an external OH scavenger injector to
measure OH concentrations using an alternative background signal, $OH_{CHEM}$, alongside the
conventional method of measuring OH using a background signal determined by tuning the
laser wavelength off-resonant to the transition, $OH_{WAVE}$ (Woodward-Massey et al.,
2020;Novelli et al., 2014;Mao et al., 2012). Intercomparison campaigns (e.g. Schlosser et al.
(2009), Onel et al. (2017a)) in the controlled environment of an atmospheric chamber are useful
to identify systematic errors in different approaches, but if both methods require calibration,
the accuracy of the measurements is still compromised by uncertainties in the calibration
methods.
In an earlier paper (Winiberg et al., 2015), accurate calibration of a FAGE instrument over
a range of external inlet pressures (440 – 1000 mbar) was performed in the Leeds HIRAC
(Highly Instrumented Reactor for Atmospheric Chemistry (Glowacki et al., 2007)) chamber.
The instrument sensitivity to OH and $HO_2$ agreed well for the conventional water vapour
calibration method (where the external pressure is always 1 bar, and external pressure effects
were simulated by altering the pressure in the FAGE detection cell) and alternative methods
based on the temporal decay of a hydrocarbon (for OH) or the temporal decay of $HO_2$ via its
second-order self-reaction (for $HO_2$) over an external pressure range of 300 – 1000 mbar. For
OH, the calibration factor, $C_{OH}$, (where $S_{HOx} = C_{HOx}.[HOx]$ and $S_{HOx}$ is the FAGE signal)
increased by 17% and for $HO_2$ a slightly greater increase in $C_{HO2}$ of 32% was determined as
the pressure increased from 350 to 1000 mbar. There was good agreement between the absolute
values and their pressure dependence for both calibration methods. Such comparisons are
particularly relevant to aircraft operation where external pressures will vary considerably
during the flight or for evacuable chambers such as the Leeds HIRAC chamber which can
operate from 50 – 1000 mbar. Marno et al. (2020) have also developed the All Pressure
Altitude-based Calibrator of HOx Experimentation (APACHE) to allow calibration of their
FAGE instrument HORUS (HydrOxyl Radical measurement Unit based on fluorescence
Spectroscopy) as a function of pressure, but not temperature.
Little is known on the effect of gas temperature at the inlet upon instrument sensitivity for
LIF instruments, despite field instruments being used at extremes of temperature, from day to
night, from deserts to the polar regions, and in aircraft, where temperatures change rapidly with
altitude. Additionally, ambient conditions influence not only the inlet temperature, but the
whole apparatus. For example in the FAGE system associated with HIRAC, based on a design
for aircraft use (Commane et al., 2010), the whole inlet tube (~30 cm) is located inside the



HIRAC chamber and so wall loss rates of HOx in the inlet tube will be influenced by the
temperature of the HIRAC chamber. The long inlet is required either to locate the pinhole
outside of the aircraft for the airborne instrument, or to allow sampling across the diameter of
the HIRAC chamber. To date, the only study investigating the effect of inlet temperature on
instrument sensitivity to $HO_x$ radicals has been performed by Regelin et al. (2013), who
reported a minor positive dependence of the OH sensitivity ($C_{OH}$) as a function of decreasing
inlet temperature for the HORUS instrument (possibly due to a cooling effect on the
instrumentation). There was a more marked decrease in the instrument sensitivity to $HO_2$ with
decreasing temperature, most probably due to enhanced wall losses at lower temperatures.
In this paper, instrument sensitivity as a function of external inlet temperature has been
determined for the HIRAC FAGE instrument for both OH and $HO_2$, using the water vapour
photolysis calibration method in an external flowtube (termed 'conventional method') and
alternative calibration methods using chemical reactions in the HIRAC chamber (Winiberg et
al., 2015) at varying temperatures. Alternative OH calibrations used the inferred [OH] from the
measured decay of a hydrocarbon (HC), typically cyclohexane, reacting with OH (R1) (termed
'HC decay method'). The rate of loss of HC is then given by equation (E1).

17         OH + HC → products         (R1)

18         $\frac{-d[\mathrm{HC}]}{dt} = k_{bi}[\mathrm{OH}][\mathrm{HC}]$         (E1)

In E(1), $k_{bi}$ is the well-established literature value for the bimolecular rate coefficient between
OH and the monitored hydrocarbon and $\frac{-d[\mathrm{HC}]}{dt}$ can be measured from the HC time series so
that [OH] is the only unknown parameter and can be calculated and compared with the [OH]
predicted via the conventional calibration method.
$HO_2$ was also calibrated by monitoring the $HO_2$ kinetic decay during the recombination
following generation by HCHO photolysis in the presence of $O_2$ (termed '$HO_2$ self-reaction
method').

26         HCHO + hν → H + HCO         (R2)

27         HCO + $O_2$ → $HO_2$ + CO         (R3)

28         H + $O_2$ + M → $HO_2$ + M         (R4)

29         $HO_2$ + $HO_2$ (+M) → $H_2O_2$ + $O_2$ (+M)         (R5)





The time dependence of the [HO$_2$] in the second-order decay depends on the initial
concentration of HO$_2$ allowing for calibration.
**2    Experimental**
**2.1    The HIRAC chamber**
The alternative calibration methods of monitoring hydrocarbon or HO$_2$ decays were conducted
in HIRAC using very similar methods and conditions as described in Winiberg et al. (2015).
HIRAC is a stainless steel chamber with a total volume of 2.25 m$^3$ and can operate over a wide
range of pressures (50 – 1000 mbar) and temperatures (227 – 343 K). Multiple access ports are
available to connect an array of instrumentation and monitoring equipment (pressure gauges,
thermocouples etc.). The chamber has been described previously in detail in Glowacki et al.
(2007), Malkin et al. (2010) and (Bejan et al., 2018). More recently a temperature control
system was installed to further enhance the capabilities of the HIRAC chamber (Section 2.1.1).
Details on the temperature characteristics of HIRAC can be found in Section S1 of the SI.
The photolysis lamps, housed in eight quartz tubes mounted radially inside the reactive
volume, were used to initiate photochemistry. The lamps were interchangeable depending on
the target molecules; lamps, with primary emissions centred at 254 and 310 nm (GE Optica,
GE55T8/HO and Philips, TL40W/12 RS respectively), were used for the alternative OH and
HO$_2$ calibration methods respectively (sections 3.2 and 3.3). The housings were flushed with
dry N$_2$ (~3 slm per housing) to help regulate the temperature and remove photolabile species
and water, which could condense or freeze around the lamps at lower temperatures. A
photolysis lamp induced chamber temperature increase of ~2 – 5 K was seen over the course
of a typical experiment (<40 mins), but this variation was reduced if the chamber was
temperature controlled. Temperatures were monitored using a series of K-type thermocouples
inside the lamp housings (one per lamp) as well as distributed around the inside of the chamber.
Thermocouples were placed strategically to allow the temperature to be measured close to the
chamber walls, inlets, flanges and in the chamber.
2.1.1  Temperature Control System
During manufacture, square cross section steel tubing (volume ~50 L) was welded directly to
the outer skin of HIRAC, allowing a cooling/heating liquid to flow around the chamber,





controlling the temperature inside. The square tubing enabled the temperature control liquid to
transfer heat more efficiently to the chamber by offering a larger contact surface area compared
to cylindrical tubing. A Huber thermostat unit (model 690W) was used to circulate ~60 L of
thermofluid (Huber DW-THERM, 183 - 473 K) around the chamber. Further details are given
in the SI (Section S1).

6       HIRAC was able to sustain a steady temperature (±2 K) across the chamber at any

temperature between 227 and 343 K and example temperature profiles are given in the SI
(Figure S2). A negligible temperature gradient was observed across the central portion of the
chamber, in both the horizontal and vertical axes. Close to the walls of the chamber, however,
a change of ~1 K was observed. The flanges around the HIRAC chamber were insulated with
~40 mm of neoprene, however there was no direct temperature control of the flanges or access
ports, which was likely responsible for the change in temperature at the large 600 mm access
flanges.
### 2.1.2   $HO_x$ Instrumentation
The OH and $HO_2$ radicals were detected using a FAGE instrument based in the HIRAC
chamber with a 5 kHz pulse repetition frequency (PRF) laser light source, as described in
Winiberg et al. (2015);Winiberg et al. (2016) and Glowacki et al. (2007). Air was sampled at
~6 slm through a 1.0 mm diameter pinhole nozzle and passed down the inlet (length 280 mm,
50 mm diameter) into the OH detection axis maintained at low pressure (typically ~3.85 mbar)
using a high-capacity rotary-backed roots blower pumping system (Leybold, Trivac D40B and
Ruvac WAU251). The long inlet was used to draw a sample away from the chamber walls
where radical losses increase (a maximum of 15% decrease at <10 mm from the chamber wall)
and to probe any radical gradients occurring due to spatially inhomogeneous production
(Winiberg et al., 2015). The FAGE instrument was coupled to the HIRAC chamber using ISO-
K160 flanges, ensuring the pinhole is kept >200 mm from the chamber walls.
Concentrations of $HO_2$ were measured simultaneously in a second detection axis ~300 mm
downstream of the OH detection axis. High purity NO (BOC, N2.5 Nitric Oxide) was added
~20 mm before the $HO_2$ detection axis into the centre of the FAGE cell in the direction of gas
flow through 1/8” stainless steel tubing at a rate of 5 sccm (Brooks 5850S) converting $HO_2$ to
OH. Conversion of some types of $RO_2$ radicals (in particular β-hydroxyperoxy radicals) to OH
upon reaction with NO has been reported in other FAGE instruments (Whalley et al.,
2013;Fuchs et al., 2011). However, during the alternative $HO_2$ calibrations (based on HCHO


photolysis) presented here no β-hydroxyperoxy radicals were generated hence any interference
was assumed to be negligible.
A JDSU Nd:YAG pumped Sirah Cobra Stretch system (PRF = 5 kHz) was used to generate
the frequency doubled ~308 nm (307.99 nm to excite the $Q_1(2)$ rotational state) light for the
fluorescence of OH radicals. Light was directed from the output of the laser and focussed into
fibre optic cables (10 m, Oz Optics) which were then attached directly to the FAGE cell arms
*via* collimators (Oz Optics). Fluctuations in laser power were accounted for using a linear
response UV sensitive photodiode (UDT-555UV, Laser Components UK) at the exit arm of
the OH and $HO_2$ detection axes to normalise the LIF signal. The laser system provided between
5 – 7 and 2 – 3 mW of 308 nm light to the OH and $HO_2$ detection axes, respectively.
The OH fluorescence was collected orthogonal to the gas flow onto electronically gated
Channeltron PhotoMultiplier tubes (CPM, Perkin Elmer, C943P) *via* a series of imaging lenses
and a narrow bandpass filter (Barr Associates, 308.8 ± 5.0 nm). A spherical concave back
reflector was positioned underneath the cell, opposite the detection optics, to optimise light
collection onto the CPM. To avoid detector saturation, the CPM was gated (i.e. switched off)
for the duration of the laser pulse using a modified gating unit based on the original design by
Creasey et al. (1997a). Signals from the CPM were analysed using PC-based photon counting
cards (Becker and Hickl PMS-400A).
### 2.1.3  Other instrumentation
As with the previously published work (Winiberg et al., 2015), a chemiluminescence $NO_x$
analyser (TEC 42C, limit of detection = 50 pptv at 60 s averaging) was used to determine that
levels of $NO_x$ (NO + $NO_2$) in the HIRAC chamber were typically below the detection limit of
the apparatus.
Most of the OH calibration experiments using the hydrocarbon decay method were
performed monitoring HC decays using a chemical ionization time of flight mass spectrometer
(Kore custom build) operating with $N_2^+$ ionization. Gas was sampled from HIRAC via ~7 m of
1/8" Teflon tubing with the inlet being located close (within 70 cm) to the FAGE inlet. A
majority of the experiments were carried out with cyclohexane as the HC (monitored at *m/z* =
84.15), although other compounds were used. The mass spectrometer signal was calibrated by
introducing known HC concentrations into HIRAC. An example of the resulting calibration
plot can be found in the SI (Section S2, Figure S3).



## 2.2 General Chamber preparation

Calibration experiments were conducted at 1000 mbar in an Ultra-High Purity (UHP) 1:4 synthetic air mix of $O_2$ (BOC, zero-grade, >99.999%) and $N_2$ (BOC, zero-grade, >99.998%) to match the range of pressures from the water vapour calibration method (section 3.1). Thorough mixing of reaction mixtures within HIRAC was achieved in ≤70 s by four circulation fans mounted in pairs at each end of the chamber. The chamber was evacuated to ~0.05 mbar for ~60 - 120 min following each experiment using the rotary pump backed roots blower to ensure removal of all reactants/products. The combined sampling rate of ~9 slm from the chamber required a counter flow of synthetic air to maintain the desired pressure and resulted in a first order dilution term of $(4.5 \pm 0.2) \times 10^{-5}$ s$^{-1}$. The dilution flow was regulated using two Brooks mass flow controllers ($N_2$ and $O_2$) and the dilution was taken in account in all analyses.

## 2.3 Chemical reagents

Known concentrations of precursors (except $H_2O_2$) and reagents were introduced to the chamber in the vapour phase through a 0.97 L stainless steel delivery vessel. Hydrogen peroxide (50% wt solution, Merck, used as supplied) was directly injected via a syringe. Multiple injections could be made in each run to ensure a wide range of [OH] was covered.

For the hydrocarbon based OH calibration method, cyclohexane (99%, Fischer Scientific), methylcyclohexane (>99.9%, Sigma Aldrich) and heptane (99%, Fischer Scientific) were purified using freeze-pump-thaw cycles before being introduced into the HIRAC chamber.

For the second-order $HO_2$ calibration method, formaldehyde (HCHO) was produced in the gas phase by gently heating paraformaldehyde (99.9%, Sigma Aldrich) into the evacuated delivery vessel. This method was sufficient for producing the 2 – 3 ppmv concentrations of HCHO in the HIRAC chamber that were required.

## 3 Calibration methods

### 3.1 Flowtube/Water Photolysis Calibration Method

The flowtube calibration method relies on the photolysis of $H_2O$ vapour at 184.9 nm in a fast flow (40 slm) of synthetic air. A mercury penray lamp (LOT-Oriel, Hg-Ar) was used as the photolysis source, placed at the end of a square cross section flow tube ($12.7 \times 12.7 \times 300$ mm). Air was humidified by passing a fraction of the bulk air flow through a bubbler containing





deionised water. The [$H_2O$] was measured using a dew-point hygrometer (CR4, Buck Research
Instrument) prior to the flow tube and the resulting OH and $HO_2$ concentrations from photolysis
can be calculated from equation (E2):

4            $[OH] = [HO_2] = [H_2O]\ \sigma_{H_2O,\ 184.9\ nm}\ \Phi_{OH}\ F_{184.9\ nm}\ \Delta t$          (E2)

where $\sigma_{H_2O,\ 184.9\ nm}$ is the known absorption cross-section of $H_2O$ vapour at 184.9 nm
(($7.22 \pm 0.22) \times 10^{-20}$ $cm^2$ molecule$^{-1}$ (Cantrell et al., 1997;Creasey et al., 2000;Hofzumahaus
et al., 1997)), $\Phi_{OH}$ ($= \Phi_{HO_2} = 1$) is the photodissociation quantum yield of OH and $HO_2$ (Fuchs
et al., 2011), $F_{184.9\ nm}$ is the photon flux of 184.9 nm light and $\Delta t$ is the exposure time of the air
to the Hg lamp output. The exposure time of the air to the 184.9 nm light, $\Delta t$, was calculated
as a function of the known velocity of the air and the cross section of the photolysis region.
The product $F_{184.9\ nm} \times \Delta t$ was determined for lamp supply currents between 0.2 and 3.0 mA
using the $N_2O$ actinometry method described in detail in a number of publications (Edwards et
al., 2003;Heard and Pilling, 2003;Faloona et al., 2004;Whalley et al., 2007;Glowacki et al.,

14    2007).

15        The gas output from the flow tube was directed towards the FAGE sampling inlet, where

the overfill of the FAGE sample volume from the flow tube stopped the impingement of
ambient air. A range of $HO_x$ concentrations ($10^8 - 10^{10}$ molecule $cm^{-3}$) were produced by
changing the mercury lamp photon flux whilst keeping a constant [$H_2O$] (typically 2000 - 3000
ppmv). The average calculated [$HO_x$] values are compared to their concurrent OH/$HO_2$ signals
observed during the same time period, the linear regression of which gives the instrument
sensitivity to OH/$HO_2$. A typical calibration plot is shown in Figure 1. Potential systematic
errors in the flowtube calibration method have been discussed previously (Winiberg et al. 2015)
and are summarized for the current instrument in Table 4 and discussed further in the SI,
Section S3, which also contains a schematic of the flowtube calibration apparatus (Figure S4).



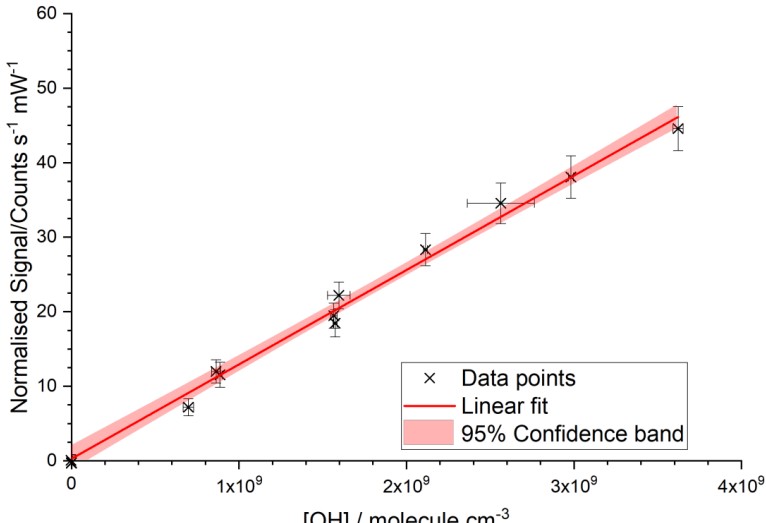

**Figure 1:** Typical room temperature calibration plot from the conventional water photolysis, flow tube
method. The total flow rate was 40 slm, with $[H_2O]$ = 1600 ppmv, the laser power was 9.65 mW and
the OH cell was at a pressure of 2.6 Torr. Gradient = $(1.266 \pm 0.034) \times 10^{-8}$ counts $s^{-1}$ $mW^{-1}$ $cm^3$
molecule$^{-1}$, intercept = $0.28 \pm 0.74$ counts $s^{-1}$ $mW^{-1}$. Errors are $2\sigma$.
### 3.1.1 Calibration for External Inlet Temperature
The FAGE inlet was wrapped with ¼" copper tubing ($\sim$ 5 cm between coils) and covered in
two layers of aluminium foil to aid thermal contact. A final layer of 10 mm thick neoprene was
added to the outside of the foil to aid insulation. The Huber temperature control unit was used
to flow DW-THERM thermofluid through the tubing to vary the temperature of the inlet.
Temperatures were monitored externally using three K-type thermocouples; two positioned on
the inlet and one on the conical pinhole nozzle during the calibration procedure (see Figure
2(a)).
Calibrations were conducted at five external inlet temperatures from 263 – 343 K,
representative of the operating temperature range for the HIRAC chamber. During the bulk of
the experiments, gases from the flowtube calibration source were maintained at room
temperature. However, an additional range of calibration experiments were performed with
flowtube gas maintained to within $\pm$5 K of the measured external inlet temperature. This effect
was achieved by passing the humidified bulk flow through a 2 m long coil of ¼" copper tubing
held at the desired set point using a thermostat controlled water bath (Thermo Fischer Science).





The $[H_2O]_{vap}$ was determined just before the calibration flowtube, with the temperature
monitored both before and at the exit of the flowtube. Short gas lines were used between the
water bath and the flow tube, which was covered in a thin layer of neoprene to insulate and
reduce temperature gradients.
**(a)**

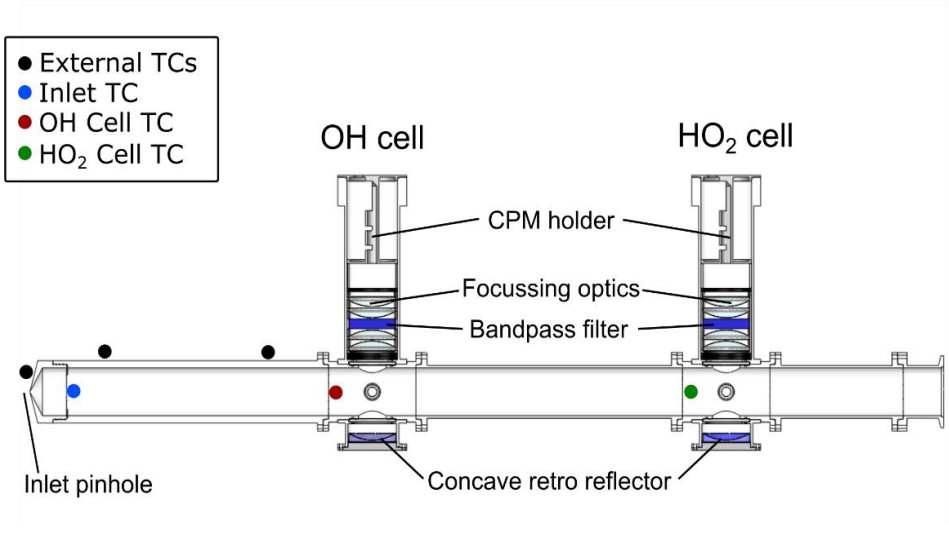

**(b)**                                          **(c)**

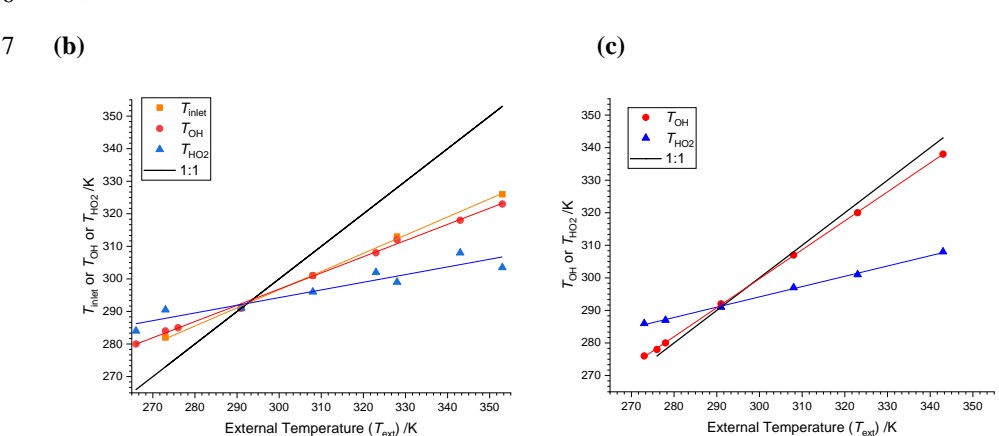

**Figure 2:** (a) Schematic of FAGE Cell showing locations of thermocouples. (b) Internal cell
temperatures ($T_{OH}$ or $T_{HO2}$) and inlet temperatures ($T_{inlet}$) plotted as a function of the external temperature
($T_{ext}$), when sampling air at 293 K from the calibration flowtube. (c) Internal temperatures as a function
of the external temperature when either sampling temperature controlled air from the calibration
flowtube or sampling from the HIRAC chamber.



Prior to the calibration, the internal cell temperatures were measured using three K-type
thermocouples positioned in the centre of the gas flow inside the inlet (just after the inlet
pinhole), OH and $HO_2$ fluorescence cells, details of which are discussed in the results section
(4.1.1). The thermocouples were inserted into the cell using a ¼" compression fitting port, seal;
this allowed the cell to be operated at normal operating pressure during the temperature profile
measurements. Thermocouples were held in place temporarily using electrical tape, and
OH/$HO_2$ calibrations were not performed with the thermocouples in place.

## 9    3.2   Hydrocarbon decay method

A majority of the hydrocarbon decay OH measurements were made with cyclohexane as the
monitored hydrocarbon (HC) (monitored via the $m/z$ = 84.15 peak) and hydrogen peroxide
photolysis at 254 nm as the OH source.
The principle of the hydrocarbon decay method was outlined in the introduction; the rate of
loss of the HC by OH is given by:
$$-\frac{d[\text{HC}]}{dt} = k_{bi}[\text{OH}][\text{HC}] \tag{E1}$$
The rate coefficient for cyclohexane, c-$C_6H_{12}$, has received much attention in the literature over
the 273 – 343 K temperature range used in this study, and so we use the IUPAC recommended
rate expression (Atkinson et al., 2006):
$$k_{\text{OH+c-C}_6\text{H}_{12}} = 3.26 \times 10^{-17}\, T^2\, e^{((262\pm66)/T)}\, \text{cm}^3\, \text{molecule}^{-1}\, \text{s}^{-1} \tag{E3}$$
The calculated [OH] from the hydrocarbon decay can be compared to the corresponding FAGE
signal, corrected for the difference in [$H_2O$] used in the calibration and that present in the
HIRAC chamber, to determine the $C_{\text{OH}}$. In practice, the total HC decay is a combination of
reaction with OH and other first order loss processes, primarily dilution (as sampled gas is
replenished with air). Therefore
$$-\frac{d[\text{HC}]}{dt} = k_{1st}[\text{HC}] + k_{bi}[\text{OH}][\text{HC}] \tag{E4}$$
where $k_{1st}$ represents the rate coefficient for the sum of all non-OH first order loss processes.
Gradients were obtained from analysis within the Origin software package. A second order
polynomial was fitted to 10 – 40 points (with the separation of each point being 10 s); the
number of points depending on the rate of change of the [HC] and the data points were
smoothed via the method of Savitzky-Golay (Savitzky and Golay, 1964).





$k_{1st}$ was determined from the HC decays in the absence of OH (either with no lamps on, or
no OH precursor present). For each injection of HC (typical initial concentration of $3 - 5 \times$
$10^{13}$ molecule cm$^{-3}$) there were multiple $H_2O_2$ injections (~1 ml). FAGE measurements were
typically averaged over 30 s (30 data points, with each data point corresponding to accumulated
signal over ~1 s) to counteract the noise arising in fluorescence counts. During rapid changes
in the observed signal, for example immediately after initial photolysis of hydrogen peroxide
in the chamber (see Figure 3(a)), a reduced averaging period was used.
Figure 3(a) shows a typical time series of OH with the black line giving the [OH] derived
from the mass spectrometer measurements and the brown line giving [OH] derived from the
FAGE signal and converted to [OH] using the conventional flow tube water vapour photolysis
calibration at 293 K. Figure 3(b) shows the resulting scatter plot. The slope of the scatter plot
gives the correction to be applied to $C_{293\,K}$ from the conventional calibration to match the [OH]
derived from the mass spectrometric measurements.

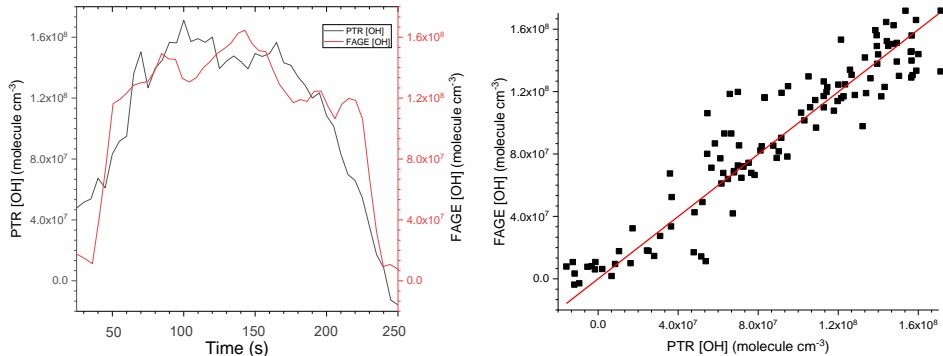

**Figure 3:** a) Time series of [OH] derived from FAGE measurements and from mass spectrometric
measurements of cyclohexane removal recorded following $H_2O_2$ photolysis at 293 K and 1000 mbar
air. b) Resultant scatter plot where the gradient, $0.998 \pm 0.016$ (2σ) gives $C_{rel}$ for the FAGE apparatus
at 293 K for this experiment. The average gradient at 293 K is $1.034 \pm 0.0068$ from five experiments.
**3.3   Calibration of HO$_2$ detection via HO$_2$ recombination kinetics**
The HCHO photolysis/HO$_2$ recombination kinetics method of HO$_2$ cell calibration was used
as described in Winiberg et al. (2015). Formaldehyde was introduced in a flow of nitrogen into
the chamber (containing synthetic air at 1000 mbar) at concentrations of
~$2 \times 10^{13}$ molecule cm$^{-3}$. The chamber was irradiated (lamps: Philips TL40W/12 RS) resulting
in an almost instantaneous HO$_2$ signal (reactions R2 – R4). Once a steady state HO$_2$





concentration was achieved, the photolysis lamps were turned off and the decay of $HO_2$ was
monitored by FAGE for ~120 s (Figure 4). The decay of $HO_2$ was primarily controlled by the
self-reaction (R5), but there was a small first-order contribution from loss to the walls (R6).
The measurement of $HO_2$ decays was repeated up to six times before the laser wavelength was
scanned to the offline position.
$$HO_2 + HO_2 \,(+M) \rightarrow H_2O_2 + O_2 \,(+M) \qquad\qquad (R5)$$
$$HO_2 \rightarrow \text{loss} \,(k_{\text{loss}}) \qquad\qquad (R6)$$
The chamber mixing fans were used for the first three calibration decays, representative of
a typical experimental homogeneous gas mixture. The second series of three calibration decays
were conducted without the mixing fans to probe the $HO_2$ recombination and wall loss kinetics
in the absence of effective mixing.
When the fans are on, the loss of $HO_2$ was characterised by bimolecular self-reactions and
a first order wall loss parameter. The solution to this mixed order decay is given by:
$$\left(S_{HO_2}\right)_t = \left( \left( \frac{1}{\left(S_{HO_2}\right)_0} + \frac{2 \cdot k_{HO_2 + HO_2}}{k_{\text{loss}} \cdot C_{HO_2}} \right) \cdot e^{(k_{\text{loss}}t)} - \left( \frac{2 \cdot k_{HO_2 + HO_2}}{k_{\text{loss}} \cdot C_{HO_2}} \right) \right)^{-1} \qquad (E5)$$
where $(S_{HO_2})_t$ and $(S_{HO_2})_0$ are the $HO_2$ signal at time $t$ and $t = 0$ respectively, $(C_{HO_2})$ is the
instrument sensitivity, $k_{HO_2 + HO_2}$ is the $HO_2$ recombination rate coefficient and $k_{\text{loss}}$ represents
the wall loss parameter. Both $k_{\text{loss}}$ and $C_{HO_2}$ were determined by data fitting the $S_{HO_2}$ decay
using equation (E5) with a Levenburg-Marquardt non-linear least squares algorithm, fixing the
initial signal and $k_{HO_2 + HO_2}$. The first ~100 s of data were used, ensuring analysis after an almost
complete decay of $S_{HO_2}$. Figure 4 shows an example of a typical decay and the resulting fit to
equation (E5).
For the experimental temperature range (275 – 345 K), $k_{HO_2 + HO_2}$ has values between
$(2.00 - 2.85) \times 10^{-12}$ $cm^3$ molecule$^{-1}$ s$^{-1}$ according to the recommendation given by IUPAC
(2007). The chamber was operated under dry conditions (< 10 ppmv $[H_2O]_{\text{vap}}$), and so the
enhancement of $k_{HO_2 + HO_2}$ by formation of a pre-reactive complex with $H_2O$ was ignored for
these analyses. The wall loss rate, $k_{\text{loss}}$, was dependent on daily chamber conditions and was
therefore determined as part of the fitting procedure along with $C_{HO_2}$, typically between
$0.032 - 0.073$ s$^{-1}$ with an uncertainty of $\pm10$ % ($2\sigma$).



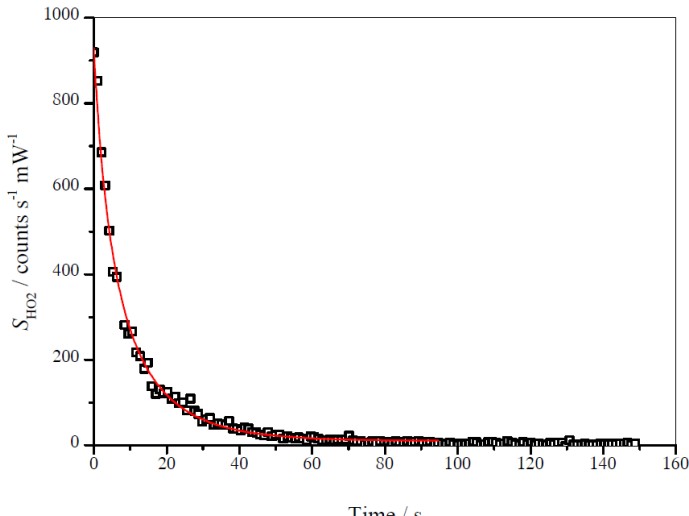

**Figure 4:** Typical $HO_2$ decay recorded at 293 K and 1000 mbar air. The red line is the fit to
the data from equation (E5) giving $C_{HO2,\ 293\ K} = (4.17 \pm 1.66) \times 10^{-8}$ counts $cm^3$ molecule$^{-1}$
$mW^{-1}$ $s^{-1}$

## 4 Results and Discussion

## 4.1 Conventional Calibration method

### 4.1.1 Temperature profiles in the FAGE instrument

Temperatures within the FAGE instrument as a function of external temperatures are shown in
Figures 2(b) and (c) and tabulated in Table 1. For Figure 2(b) and the first part of Table 1, the
temperatures were recorded with FAGE sampling air at 293 K from the calibration flow tube
as the FAGE inlet was cooled or heated. Temperatures became closer to ambient (293 K) from
the inlet ($T_{inlet}$) to the OH observation cell ($T_{OH}$) and finally to the $HO_2$ observation cell ($T_{HO2}$).
In Figure 2(c) and the second part of Table 1, the sampled air (either from the calibration flow
tube or from HIRAC) matched the external temperature of the inlet tube. For these experiments,
there was no thermocouple located inside the inlet to give $T_{inlet}$. The temperature in the OH cell
was very close to the external temperature of the sampled air. The transmission process through
the FAGE inlet following sampling through the pinhole should be similar to when FAGE is in
HIRAC, however, even with the temperature controlled air in the wand calibration, it is still
difficult to determine the actual temperature and conditions at the pinhole itself.





The gap between the OH and $HO_2$ cells means that the sampled air was closer to ambient
room temperatures when reaching the $HO_2$ cell. $HO_2$ was predominantly be exposed to a
temperature environment similar to that for OH as it passed through the inlet, which may
influence wall loss rates. The variation in $T_{OH}$ and $T_{HO2}$ under different calibration regimes
means that care has to be taken in comparing $C_{HOx}$ values, as a number of processes within
FAGE are temperature dependent. Nevertheless, the different calibration methods do yield
important insights into the processes in the FAGE apparatus.
**Table 1**: Temperature Calibration of the FAGE instrument with a) constant temperature (293
K) calibration gas b) with calibration gas at the external temperature.

| External Temperature/K ($T_{ext}$) | Inlet Temperature /K, ($T_{inlet}$) | OH FAGE Cell Temperature /K, ($T_{OH}$) | HO$_2$ FAGE Cell Temperature /K, ($T_{HO2}$) |
|---|---|---|---|
| *Ambient Calibration Air at 293 K* | | | |
| 266[a] | | 280 | 284 |
| 273 | 282 | 284 | 290.5 |
| 276 | | 285 | |
| 293 | 293 | 293 | 293 |
| 308 | 301 | 301 | 296 |
| 323 | | 308 | 302 |
| 328 | 313 | 312 | 299 |
| 343 | | 318 | 308 |
| 353 | 326 | 323 | 313.5 |
| *Calibration Air Matched to FAGE Inlet Tube Temperature* | | | |
| 273 | | 276 | 286 |
| 276 | | 278 | |
| 278 | | 280 | 287 |
| 293 | | 293 | 293 |
| 308 | | 307 | 297 |
| 323 | | 320 | 301 |
| 343 | | 338 | 308 |

a – All temperature measurements have uncertainty of ± 0.5 K.
Figures 2(b) and (c) show the linear relationship between the internally measured
temperature at the pinhole, OH cell and $HO_2$ cell. For Figure 2(b), the linear regression of the
data gives ratios of $0.556 \pm 0.002$, $0.510 \pm 0.002$ and $0.195 \pm 0.002$ for the inlet thermocouple
(close to the pinhole), OH cell and $HO_2$ cell. The temperature in the OH cell is controlled by
the external temperature. In contrast, in field instruments which have a very different design
and where OH is probed very close to the pinhole, there is a significant cooling effect due to





the expansion (Creasey et al., 1997b). This is lost in the HIRAC FAGE due to the long inlet
prior to probing the OH.

### 4.1.2  Temperature Dependent Flow Tube Calibration with Air at 293 K

Figure 5 displays the relative $C_{OH}$ and $C_{HO_2}$ for the HIRAC FAGE instrument as a function of
external temperature between 266 – 343 K, with the data points listed in the top half of Table
2. In these experiments the FAGE inlet was cooled or warmed to give the external temperature
($T_{ext}$). The air from the calibration flow tube was at a constant 293 K and therefore the
temperature in the observation cells (OH or $HO_2$) was varying compared to the inlet air. This
method of investigating the temperature dependence of $C_{HOx}$ therefore operates under different
conditions from the subsequent methods (Sections 4.1.3 and 4.2). Data for $C_{HOx}$ are presented
relative to the calibration factor at room temperature (293 K).

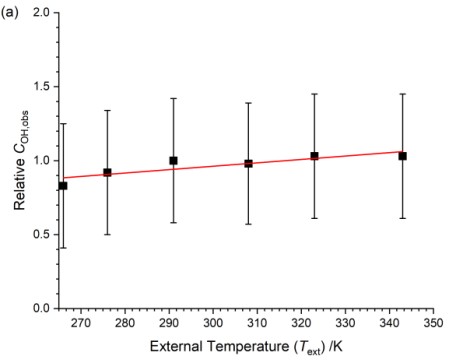
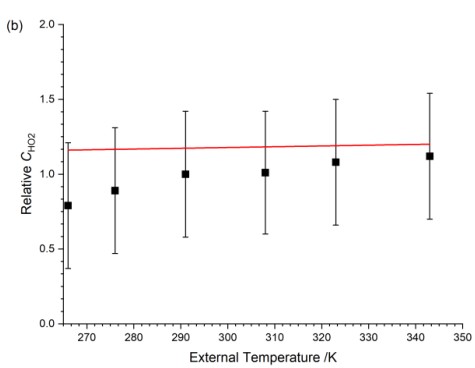

**Figure 5:** Temperature dependence of the calibration factors ($C_{HOx}$) as a function of the external
temperature with HOx being delivered from the calibration flow tube at a constant temperature. (a)
$C_{OH,obs}$, slope = $(0.0023 \pm 0.0007)$ K$^{-1}$. (b) $C_{HO2,obs}$, slope = $(0.0005 \pm 0.0031)$ K$^{-1}$. Errors are 2σ.
$C_{OH,obs}$ shows a positive temperature dependence $(0.0023 \pm 0.0007$ K$^{-1})$, for $C_{HO2,obs}$, the data
appear to be more scattered and no systematic trend is observable. The overall temperature
dependence of both HOx calibration factors are small compared to the overall uncertainty in
the calibration (40 %); the relative calibration factor for OH changes by about 20 % from 266 –
343 K. However, the error bars in Figure 5 represent the total error in the calibration, much of
which will be temperature independent. A full discussion on the temperature dependence of
the calibration factors is presented in Section 4.3.
**Table 2**: Instrument sensitivity to OH, $C_{OH}$, and HO$_2$, $C_{HO2}$, determined using the
conventional water vapour calibration method.

| $T_{ext}$/K | $T_{OH}$ / K | $T_{HO2}$ / K | $C_{OH,obs}$ | $C_{HO2,obs}$ |
|---|---|---|---|---|
| *Ambient Calibration Air at 293 K* | | | | |
| 266 | 280 | 284 | $0.83 \pm 0.42$ | $1.11 \pm 0.26$ |
| 276 | 285 | - | $0.92 \pm 0.42$ | -[a] |
| 293 | 293 | 293 | $1.00 \pm 0.42$ | $1.00 \pm 0.50$ |
| 308 | 301 | 297 | $0.98 \pm 0.41$ | $1.36 \pm 0.31$ |
| 323 | 308 | 302 | $1.03 \pm 0.42$ | $1.40 \pm 0.38$ |
| 343 | 318 | 308 | $1.03 \pm 0.42$ | $1.01 \pm 0.32$ |
| *Calibration Air Matched to FAGE Inlet Temperature ($T_{in}$)* | | | | |
| 276 | 278 | - | $1.06 \pm 0.39$ | -[a] |
| 278 | 280 | 287 | $0.91 \pm 0.50$ | $1.43 \pm 0.54$ |
| 293 | 293 | 293 | $1.00 \pm 0.40$ | $1.00 \pm 0.45$ |
| 323 | 320 | 301 | $1.18 \pm 0.39$ | $1.91 \pm 0.38$ |
| 343 | 338 | - | $1.45 \pm 0.39$ | -[a] |

The internal temperatures ($\pm 0.5$ K) for the OH and HO$_2$ fluorescence cells are represented by $T_{OH}$ and
$T_{HO2}$ respectively. a – determination of $C_{HO2}$ was precluded by a malfunctioning NO mass flow
controller.
4.1.3 Temperature Dependent Flow Tube Calibration with Air at Varying Inlet
8       Temperatures

A similar procedure to Section 4.1.2 was carried out, but in this case, the air flowing into the
calibration flow tube had been cooled/heated to match the external temperature of the FAGE
inlet. This method will give conditions that are more closely matched to those when the FAGE
instrument is located in the HIRAC chamber, where the FAGE inlet is at the same temperature
as the gas being sampled from HIRAC. The water vapour concentration was measured at a
fixed temperature in the dew-point hydrometer and therefore the [HOx] emitted from the wand
needed to be corrected for the change in [H$_2$O] and additionally, for the change in $\Delta t$ in equation
(E2).
In this calibration arrangement the temperature of the OH cell ($T_{OH}$) was virtually identical
to the external temperature ($T_{ext}$). The HO$_2$ FAGE cell was closer to ambient room temperature.
The temperature dependence of $C_{HOx,obs}$ relative to 293 K is shown in Figure 6. The calibrations
were taken at different times from those in Section 4.1.2, but the absolute $C_{HOx}$ factors at 293
K were in good agreement, within 5%. For OH, the slope of Figure 6(a) is again positive. For
HO$_2$ (Fig 6(b)) there are only three datum points and they are somewhat scattered.



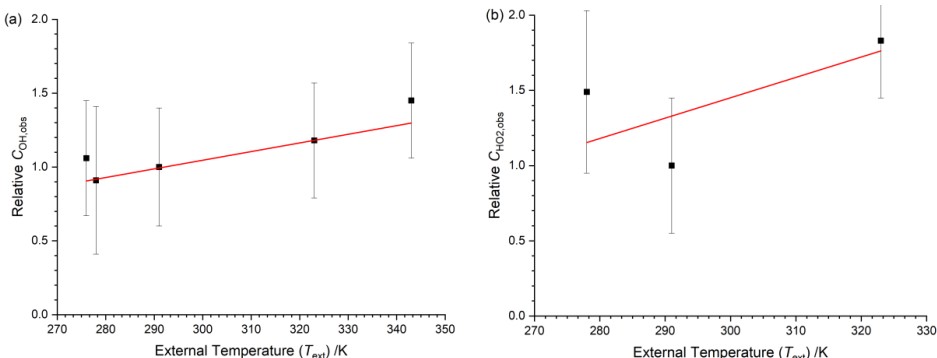

**Figure 6:** Temperature dependence of the calibration factors ($C_{HOx,obs}$) as a function of the external temperature with HOx being delivered from the calibration flow tube at the external temperature. (a) $C_{OH,obs}$, slope = $(0.0059 \pm 0.0015)$ K$^{-1}$. (b) $C_{HO2,obs}$, slope = $(0.014 \pm 0.013)$ K$^{-1}$.

## 4.2 Alternative Calibration Methods

### 4.2.1 Hydrocarbon Decay Calibration of OH Sensitivity

The ratio of the conventional water vapour flowtube calibration to the HC decay method derived from scatter plots such as Figure 3 at 293 K was $1.034 \pm 0.068$, where the errors are the statistical errors in the gradient of the scatter plots at the $2\sigma$ level. The two methods are therefore in excellent agreement as has been observed in our previous study conducted solely at room temperature (Winiberg et al. (2015), $1.19 \pm 0.26$). The increased number of data points available for the HC analysis using PTR monitoring increases the precision of this work compared to our earlier studies where [HC] was measured at much lower time resolution by FTIR or gas chromatography.

A potential source of error in the HC decay method is quantifying the removal of the HC by non-OH sources. The effects of dilution and wall loss can be accounted for by suitable blank experiments, however, it is harder to account for any other chemically induced removal by photolytically generated radicals other than OH in such blank experiments. The hydrocarbons chosen for this analysis are simple alkanes with well-established chemistry that should minimize such possibilities i.e., very slow reactions with any photolytically generated $O_3$ or $NO_3$. In addition, when both cyclohexane (CH) and heptane (HEP) were used as the HC, the gradient of the resulting relative rate plot ($\ln([HEP]_0/[HEP]_t)$ vs $\ln([CH]_0/[CH]_t)$, slope = $0.923 \pm 0.010$) was in good agreement the ratio of the literature rate coefficients for OH reactions





($k_{HEP}/k_{CH} = 0.97 \pm 0.14$ at 298 K (Atkinson, 2003)). This confirms that OH was the dominant
route for chemical removal (see SI, Section S4).
**Table 3**: Temperature Dependence of $C_{OH,obs}$ Determined via the Hydrocarbon Decay Method

| Temperature/K ($\pm$0.5 K) | $C_{OH,obs}$ relative to the HC decay method at 293 K |
|---|---|
| 273 | $0.92 \pm 0.17^{a}$ |
| 293 | $1.00 \pm 0.18$ |
| 323 | $1.10 \pm 0.20$ |
| 348 | $1.21 \pm 0.22$ |

a – errors represent the total uncertainty in $C_{OH}$, see Table 4.
Displayed in Table 3 is the instrument sensitivity to OH radicals, $C_{OH,obs}$, measured between
273 and 348 K at 1000 mbar HIRAC chamber pressure using the hydrocarbon decay method
and Figure 7(a) shows these data as a function of the HIRAC temperature. An increase in $C_{OH}$
is observed. As with the experiments carried out in Section 4.1.2, the temperature of the OH
cell ($T_{OH}$) is very close to that of the gas being sampled at the inlet.

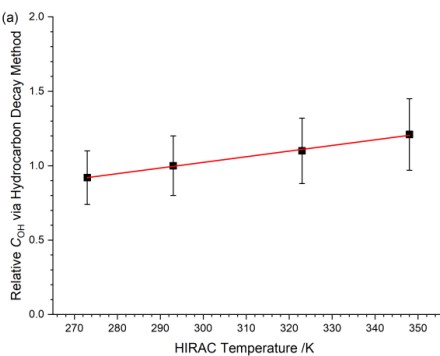
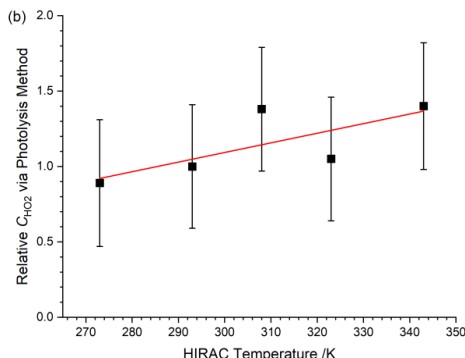

**Figure 7**: Temperature dependence of $C_{HOx,obs}$ relative to values at 293 K. (a) Relative $C_{OH,obs}$ from the
HC decay method. Slope = $(0.0038 \pm 0.0007)$ K$^{-1}$ (b) Relative $C_{HO2,obs}$ from the HCHO photolysis
method. Slope = $(0.0064 \pm 0.0034)$ K$^{-1}$. Errors are 2σ.





**Table 4:** The systematic uncertainties in the various parameters that determine the accuracy in
the OH and $HO_2$ calibration factors for the conventional and alternative calibration methods.

| Conventional Flowtube | | Hydrocarbon Decay | | HCHO + $hv$ | |
|---|---|---|---|---|---|
| Parameter | Uncertainty | Parameter | Uncertainty | Parameter | Uncertainty |
| $F_{184.9\ nm} \times t$ | 20% | $k_{OH} - c\text{-}C_6H_{12}$ | 12%[a] | $k_{HO_2 + HO_2}$ | 38%[e] |
| $[H_2O]$ | 1% | $k_{Dil}$ | 2%[b] | $S_{HO_2}$ initial | 10%[f] |
| $\sigma_{H2O}$ | 3% | $[c\text{-}C_6H_{12}]$ | 5% | Laser power | 6% |
| Laser power | 6% | Gradient | 10% | Online Position | 4%[c] |
| Online Position | 4%[c] | Laser power | 6% | | |
| | | Online Position | 4%[c] | | |
| Error | **22%**[d] | Error | **18%**[d] | Error | **40%**[d] |

a – Error estimated from literature review. Five recent determinations (NIST Kinetics) of the 298 K rate coefficient
give ~5% spread, added some additional uncertainty to account for temperature dependence.
b – Dilution determined from flow controller measurements.
c – The online position error is the approximate error in the maximum line intensity that is achieved when
positioning the laser wavelength at the centre of the OH transition.
d – Total accuracy is taken as the sum in quadrature of the individual uncertainties.
e – Error in rate coefficient from the IUPAC evaluation.
f – Uncertainties in the fitting parameters.
Table 4 summarizes the errors associated with the alternative calibration methods. For the
hydrocarbon decay method, the major uncertainties are in the rate coefficient of the
hydrocarbon (~12% for OH + cyclohexane), determination of cyclohexane concentration (5%)
and the gradient of the cyclohexane decay (10%). Other uncertainties are drifts in the laser
power (~6%, determined from monitoring a photodiode) and wavelength position (~4%).

### 4.2.2   Calibration via $HO_2$ recombination kinetics

Displayed in Table 5 is the instrument sensitivity to $HO_2$, $C_{HO2,obs}$, determined using the
alternative calibration method between 273 and 343 K at 1000 mbar chamber pressure. Figure
7(b) shows $C_{HO2}$ as a function of temperature relative to the instrument sensitivity at 293 K.
Each measurement point represents the weighted average of at least five experimental data sets
and the error bars represent the total uncertainty in the instrument sensitivity to $\pm2\sigma$. As with
the hydrocarbon decay method, the overall uncertainty is calculated as the sum in quadrature
of fit precision to the decay and the systematic uncertainties listed in Table 4. The largest
uncertainty was in the $HO_2$ self-reaction rate coefficient, dependent on the temperature used
(38%). The slope of the linear fit to the $C_{HO2}$ values is $(0.0064 \pm 0.0034)$ K$^{-1}$. The absolute





agreement between the conventional and HCHO photolysis methods at 293 K is good with
$C_{HO2, conventional} = (3.38 \pm 1.08) \times 10^{-8}$ counts cm$^3$ molecule$^{-1}$ mW$^{-1}$ s$^{-1}$ and $C_{HO2, HCHO\ photolysis} =$
$(3.69 \pm 1.48) \times 10^{-8}$ counts cm$^3$ molecule$^{-1}$ mW$^{-1}$ s$^{-1}$.
**Table 5:** Instrument sensitivity to HO$_2$, $C_{HO2}$, determined using the HCHO photolysis method
over the 273 – 343 K external inlet temperature range.

| $T_{HIRAC}$ / K[a] | $T_{HO2}$ / K[a] | $C_{HO2}$ (rel. 293 K)[b] |
|---|---|---|
| 273 | 286 | 0.89 ± 0.36[c] |
| 293 | 293 | 1.00 ± 0.40 |
| 308 | 297 | 1.38 ± 0.55 |
| 323 | 302 | 1.05 ± 0.42 |
| 343 | 308 | 1.40 ± 0.56 |

a – Error in temperature ± 0.5 K.
b – Values are relative to $C_{HO2, 293\ K}$ of $(3.69 \pm 1.48) \times 10^{-8}$ counts cm$^3$
molecule$^{-1}$ mW$^{-1}$ s$^{-1}$.
c – Each $C_{HO2}$ represents the weighted average of at least 5 individual
determinations. All experiments were conducted in 1000 mbar synthetic
air mixture.

## 4.3 Discussion of calibration methods and temperature dependence

### 4.3.1 Comparison of calibration methods

For room temperature, there is excellent agreement between the wand calibration and that for
OH based on hydrocarbon decays ([OH]$_{wand}$:[OH]$_{HC}$ = 1.00:0.97) and HO$_2$ based on HCHO
photolysis and the kinetics of the HO$_2$ recombination reaction ([HO$_2$]$_{wand}$:[ HO$_2$]$_{kinetics}$ =
1.00:1.09). This is consistent with our earlier study (Winiberg et al. 2015) and has also been
confirmed in an intercomparison in the HIRAC chamber of the FAGE and NIR – CRDS (near
infrared cavity ring down spectroscopy) for HO$_2$ (Onel et al., 2017a) and CH$_3$O$_2$ (Onel et al.,
2020;Onel et al., 2017b).
For the hydrocarbon decay method there are several advantages compared to the
conventional wand calibration:
1) The [OH] is much closer to the conditions typically used in a chamber experiment ($10^6$ –
$10^8$ molecule cm$^{-3}$) whereas the lowest [OH] used in the wand calibration performed here is
typically $10^8$ molecule cm$^{-3}$. Ideally one should calibrate over the same range as used in an
experiment.
2) This work has shown that there is a temperature dependence to the calibration factors.
Calibrating via the hydrocarbon decay method provides identical conditions (temperature





and pressure) to that of a real experiment in the HIRAC chamber. Temperature variation
can be simulated using the conventional wand device, but this introduces additional
uncertainty.
3) Conventional calibrations always take place with a significant water concentration, whereas
the water concentration in the hydrocarbon decay can be set at any value.
4) Calibration can be achieved without removing the FAGE apparatus from the HIRAC
chamber decreasing the time taken for calibration.
There are some disadvantages too. The calibration for OH is strongly dependent on the
accuracy of the HC rate coefficient. It is therefore important to use a hydrocarbon with a well-
characterised rate coefficient; realistically, even the best-characterised rate coefficient is likely
to have an uncertainly of 5 – 10%. Several HC can be used to give multiple independent
determinations of $[OH]_{HC}$, but this may increase the complexity of the analysis (e.g. coincident
mass spectral peaks, or overlapping FTIR spectra) and reduce the absolute concentration of
OH. Determination of $[OH]_{HC}$ also relies on an accurate and precise determination of the
concentration gradient and the [HC] at that time. PTR measurements provide a near continuous
output, but if the [HC] is measured using systems with lower sampling rates (e.g. FTIR or GC),
there can be a significant loss in precision of the gradient measurement.
Many of the advantages and disadvantages of the hydrocarbon decay method also apply to
$HO_2$ kinetics method for $HO_2$ calibration. The rate coefficient for $HO_2$ recombination has a
higher degree of uncertainty than many OH + hydrocarbon rate coefficients and is dependent
on the amount of water present. In the HIRAC chamber the humidity can be kept very low, but
that may not be possible in all chambers; in these circumstances the humidity would need to
be measured and the rate coefficient adjusted.
All calibration methods are subject to systematic uncertainties, the magnitude of which may
vary with conditions and therefore it is sensible to use a range of calibration methods.
### 4.3.2  Temperature dependence of $C_{HOx}$
Table 6 compares the relative observed $C_{HOx,obs}$ calibration factors for the three different
calibration methods. In all cases, a positive temperature dependence is observed, but for $C_{HO2}$,
only the alternative calibration method displays a statistically significant positive slope.
The $C_{HOx}$ factors can be broken down into temperature independent components (laser
power, solid angle of fluorescence collection, detector efficiency etc) and temperature



dependent terms. Four temperature dependent terms are relevant for $C_{HOx}$: the number density
of OH in the cell, the quenching efficiency of the fluorescence, the population of the probed
quantum state of OH and the transmission efficiency through the pinhole and inlet tube
(Creasey et al., 1997b). The first three terms can be calculated and hence accounted for. Any
residual temperature dependence of $C_{HOx}$ should then relate to the transmission coefficient
through the apparatus.
*HOx number density* – The calculated [HOx] delivered to the FAGE apparatus depends on the
temperature of the HOx source, either the wand (operating at a fixed $T = 293$ K (Method 1) or
at $T_{ext}$ (Method 2) or the HIRAC chamber. If the temperature of the HOx cells are different
from this temperature, then there will be a change in the number density of HOx, over and
above that caused by the pressure changes between the HOx source (1 bar) and the HOx cell
(typically 3.6 mbar). As the temperatures of the HOx cells have been measured it is
straightforward to correct for the different number density in the observation cells and the
resulting contribution to the temperature dependence of $C_{HOx}$ as summarized in Tables S2-4.
*Quenching* – As shown in Faloona et al. (2004), the quenching parameter, $Q(T)$, is defined by
integrating the OH fluorescence decay over the defined sample time, or gated region. The
quenching rate coefficients for $N_2$, $O_2$ and $H_2O$ have been shown to be dependent on
temperature (Copeland and Crosley (1986) and (Bailey et al., 1997) for $N_2$ and $O_2$, and Bailey
et al. (1999) for $H_2O$). The total decay intensity is defined by: $[OH(A^2\Sigma^+, v' = 0)]_0 \exp(-\Gamma t)$,
where $\Gamma$, the total OH lifetime, is defined approximately as the sum total of the radiative
lifetime for OH, $\gamma$, and the non-radiative lifetime due to quenching by the aforementioned bath
gases. Bailey et al. (1997) have calculated the impact of temperature on quenching accounting
for both the change in the quenching rate coefficients and the change in the number density of
the quenchers. Both the rate coefficient for quenching and the quencher number density
decrease with increasing temperature and hence quenching overall decreases with increasing
temperature (summarized in Table S5), enhancing the fluorescence quantum yield.
*Rotational population* – The rotational population of the probed state in the $Q_1(2)$ transition
will vary with temperature. The $Q_1(2)$ is the transition giving the largest signal between 280 –
340 K, the limits of $T_{OH}$ explored in the study. Relative to ambient temperature, the rotational
population probed by $Q_1(2)$ increases by 3.5% at 280 K and decreases by 9.0% at the highest
$T_{OH}$ of 340 K (Table S6).





It is therefore possible to calculate the expected variation in $C_{HOx}$ for the different calibration
methods dependent on OH number density, quenching and rotational population; these can be
compared with the observed variation in $C_{HOx}$ summarized in Table 6. Full details on the
temperature dependences of the above components, which vary slightly with the calibration
method used are presented in Section S5 of the SI.
The difference between the observed $C_{HOx}$ and the calculated $C_{HOx}$ due to the above
parameters is attributed to increased transmission of HOx through the pinhole and inlet tube
and is given in Table 6. The $HO_x$ transmission, to the fluorescence region will depend on the
magnitude of heterogeneous loss of radicals to the walls of the FAGE inlet. The wall loss
process is a combination of diffusion and uptake at the wall and the actual temperature
dependence will depend on the radical, conditions and wall composition (Howard, 1979).
For the OH calibrations, there is an increase in OH transmission with temperature across all
three calibration methods, consistent with a decrease in OH loss to the walls which has been
observed in previous flow tube studies. OH wall loss rate in the inlet tube is usually
approximated to a first order process with a rate coefficient, $k_w$, and decreasing values of $k_w$
with temperature have been reported for flow tube studies of OH reactions (Howard, 1979), for
example Brown et al. (1990) report $k_w$ decreasing from 35 s$^{-1}$ at 227 K to 5 s$^{-1}$ at room
temperature.
For $HO_2$ measurements, there is potentially a further temperature dependent component, the
conversion of $HO_2$ into OH via R7:

21                     $HO_2 + NO \rightarrow OH + NO_2$                             (R7)

The rate coefficient for this reaction has a negative temperature dependence and the increased
number density of NO would further enhance the rate of reaction at lower temperatures. The
experiments reported in this work operated with excess NO such that the small variations in
the rate of reaction over the range of $T_{HO2}$ (284 – 313 K) will not alter the conversion of $HO_2$
to OH. However, if one were working at lower $HO_2$ conversions to mitigate against $RO_2$ to OH
conversion (Whalley et al. 2013), then variations in the conversion efficiency could change
$C_{HO2}$ as a function of temperature.
Temperature dependent $HO_2$ calibrations based on the conventional wand method give
significant scatter, but a positive increase in $HO_2$ transmission is observed for the alternative
calibration method based on $HO_2$ kinetics, the magnitude of which is similar to that for OH,
albeit with significant error bars. In general, $HO_2$ and $RO_2$ radicals exhibit lower wall loss rate



**Table 6:** Summary of the temperature dependence of $C_{HOx}$ with different calibration methods

| Method | Observed slope of relative $C_{OH,obs}$ with temperature | Calculated contribution[a] | Difference (relative OH transmission) | Observed slope of relative $C_{HO2,obs}$ with temperature | Calculated contribution[a] | Difference (relative HO_2 transmission) |
|---|---|---|---|---|---|---|
| Heated FAGE inlet, ambient air at 293 K | $(0.0023 \pm 0.0007)$ K$^{-1}$ | $(0.0001 \pm 0.0010)$ K$^{-1}$ | $(0.0022 \pm 0.0012)$ K$^{-1}$ | $(0.0005 \pm 0.0031)$ K$^{-1}$ | $(0.0000 \pm 0.0010)$ K$^{-1}$ | $(0.0000 \pm 0.0032)$ K$^{-1}$ |
| Heated FAGE inlet, match air | $(0.0059 \pm 0.0015)$ K$^{-1}$ | $(0.0029 \pm 0.0010)$ K$^{-1}$ | $(0.0030 \pm 0.0018)$ K$^{-1}$ | $(0.014 \pm 0.013)$ K$^{-1}$ | $(0.0033 \pm 0.0010)$ K$^{-1}$ | $(0.0029 \pm 0.0016)$ K$^{-1}$ |
| Alternative kinetics based methods | $(0.0038 \pm 0.0007)$ K$^{-1}$ | $(0.0027 \pm 0.0010)$ K$^{-1}$ | $(0.0011 \pm 0.0012)$ K$^{-1}$ | $(0.0064 \pm 0.0034)$ K$^{-1}$ | $(0.0032 \pm 0.0010)$ K$^{-1}$ | $(0.0032 \pm 0.0035)$ K$^{-1}$ |

a - Contribution from the change in number density, quenching and relative rotation population in the probed state.





coefficients, but in our FAGE system, $HO_2$ molecules have to travel further to reach the
titration region where reaction occurs with NO to convert $HO_2$ to OH. Therefore, there is also
potential for OH loss from the titration point to the second detection cell.

### 4.3.3 Comparison with other instruments

The temperature dependence of the calibration factors will be strongly dependent on the design
of the FAGE apparatus. Our instrument was designed with a long (~ 1 m) inlet such that we
can probe across the diameter of the HIRAC chamber to check for radial distributions of
radicals (Malkin et al., 2010). Hence, we would expect HOx transmission to play a significant
role in the temperature dependence of the calibration factor which is observed. Any similarly
designed instrument would have a contribution from HOx transmission, the magnitude of
which would depend on inlet length/residence time and construction material. Heating the inlet
should reduce transmission losses. The aircraft based instrument, from the Juelich research
group, uses a PID controlled heater to maintain their FAGE inlet at ~300 K, mitigating any
possible temperature effects. They have an in-field calibration system, also, which has shown
negligible deviation from the expected behaviour at 300 K, based on the sample gas altitude
temperature (Marno et al., 2020).

17       Regelin et al. (2013) have reported a similar temperature dependence study of $C_{OH}$ and $C_{HO2}$

as the current flowtube study with the aircraft based HORUS instrument. Cooling lines were
wound around the inlet to simulate the measured temperature profile and ambient air was
sampled from a calibration flow tube. In contrast to our slight increase in $C_{OH}$ with temperature
in the flow tube experiment, Regelin et al. observed a slight negative dependence of the OH
signal. Regelin et al. report that their calculations have shown that the sample forms a jet
between the pinhole and the OH cell such that there is insignificant interaction with the walls
and therefore transmission will not be a problem.

25       In contrast, a significant decrease in $HO_2$ signal, $S_{HO2}$, (50%) was observed as the

temperature was decreased from ~295 to ~262 K (slope = 0.017 $K^{-1}$ normalised to $S_{HO2,293\ K}$),
i.e. the same qualitative behaviour as we observed, approximately a factor two greater than
measured in our work, based on $HO_2$ recombination kinetics. Beyond the OH cell in the
HORUS experiment, the jet breaks up and Regelin et al. suggest that temperature dependent
wall losses are responsible for the change in $S_{HO2}$. Quantitative comparisons cannot be made
due to the differences in construction. The observed temperature dependence of $C_{OH}$ and $C_{HO2}$



for the HORUS and HIRAC experiments emphasise the important of performing calibrations
for each instrument under conditions as close as possible to those used in measurements.

## 5    Conclusions

The effect of temperature of the incoming sample on the sensitivity of the HIRAC FAGE
instrument to OH and $HO_2$ has been investigated between 266 and 348 K using a combination
of conventional water vapour photolysis/flow tube method (Faloona et al.) and alternative
calibration methods based on hydrocarbon decays for OH and the $HO_2$ self-reaction for $HO_2$.
In all cases, a positive increase in sensitivity was observed (Table 6) although with large error
bars in the case of $HO_2$ with conventional calibration.

10       The temperature dependence of the calibration factor can be broken down to four

components. Variations in three parameters: number density, quenching and rotational
population of the probed level, can be accounted for if the temperature and pressure in the LIF
cells are monitored. The difference between the observed and calculated temperature
dependence for the above parameters, has been attributed to HOx transmission from the pinhole
to the relevant detection chamber.

16       The temperature dependence of $C_{HOx}$ will depend on the design and construction materials

of the FAGE apparatus. It is therefore difficult to utilise the results of this study to predict
results in other systems. However, for any systems with significant sampling inlet residence
times, such as the HIRAC FAGE described in this work, increased HOx transmission with
increasing temperature should be expected. Therefore, maintaining the inlet at a relatively high
temperature should improve sensitivity in low temperature applications.

22       The *in situ* calibration methods (hydrocarbon decay and $HO_2$ recombination kinetics) offer

important advantages in that the FAGE apparatus is calibrated under the physical conditions
and [HOx] that more closely correspond to real experiments. All calibration methods are
subject to significant uncertainty, however, the origins of these uncertainties are different and
hence good agreement between calibration methods should provide confidence that significant
systematic errors are not present.

## Supplementary Information

Supplementary information; HIRAC temperature profiles, calibrations, further discussions on
calibration uncertainties, relative rate plots to confirm OH as the key species in hydrocarbon



removal and further discussion on the temperature dependence of the FAGE signal can be
found at ******.
**Author Contributions**
FAFW and IGB led the initial work on OH temperature dependence performing all experiments
with external calibration, WJW, THS and GB completed the experiments with HC decays in
HIRAC, CAB and IGB completed experiments on $HO_2$ temperature dependence. PWS, DEH
and DS planned and supervised the experiments and wrote the manuscript with contributions
from all co-authors.
**Competing Interests**
DEH is a member of the editorial board of AMT, otherwise the authors declare that they have
no conflict of interest.
**Acknowledgements**
The authors would like to thank NERC for studentships for FAFW and WJW. CAB was
sponsored by a studentship from EPSRC. GB was supported by NERC grant NE/S010246/1,
IB by the Marie Curie Fellowship LAMUNIO (no. 302342) and THS by the EU funded
EUROCHAMP2020 project.

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
