# Peer review of "Comparison of temperature dependent calibration methods"

_Atmospheric Measurement Techniques, 2023_

## Author Comment (AC1)

**Comparison of temperature dependent calibration methods of an instrument to measure OH and HO2 radicals using laser-induced fluorescence spectroscopy**

Frank A. F. Winiberg1,2, William J. Warman1, Charlotte A. Brumby1, Graham Boustead1, Iustinian G. Bejan1,3, Thomas H. Speak1, Dwayne E. Heard1, Daniel Stone1 and Paul W. Seakins1

**Response to Reviewers**

We are grateful for the reviewers' careful reading of the manuscript, constructive comments and suggestions. The original comments are in black with our responses in blue and examples of additional text in red. Page and line numbers in our responses refer to the tracked changes document.

**Reviewer 1**

The manuscript "Comparison of temperature dependent calibration methods of an instrument to measure OH and HO2 radicals using laser-induced fluorescence spectroscopy" by Winiberg et al. describes experiments to determine the temperature dependence of FAGE calibration factors. Two different methods have been used: (i) in an external calibration using a wand with a thermostatted FAGE inlet with either room temperature or thermostatted calibration gas and (ii) a chemical method taking place in the temperature controlled HIRAC chamber: hydrocarbon decay for OH calibration and measurement of HO2 decays due to self-reaction for HO2 calibration. The work has been carried out carefully and the results are presented in a clear manner. Even though the dependence of the calibration factor on temperature depends on the design of the FAGE system, and thus the obtained results are not transferable to other FAGE systems, the work is interesting to the FAGE community and I recommend publication after taking in account some minor comments.

We are grateful to the referee for these positive comments and for the careful reading of the manuscript.

Page 2, line 30: an OH interference due to ROOOH has been identified on one FAGE instrument. This possible interference should be mentioned (Fittschen et al. ACP 19, 349-362 (2019)). Reference has been added.

p5, l22:typo Corrected.

p6, I8 : what means quantitatively negligible ? Value has been included (<0.5 K) and sentences have been slightly changed as shown below.

HIRAC was able to sustain a steady temperature  $(\pm 2 \text{ K})$  across the chamber at any temperature between 227 and 343 K and example temperature profiles are given in the SI (Figure S2). A negligible temperature gradient (<0.5 K, see Figure S2) was observed across the central portion of the chamber, in both the horizontal and vertical axes.

p8, I16: is the H2O2 used stabilized ? The stabilizer can be a source of interference in the FAGE. Did you observe such phenomenon or H2O2 unstabilized has been used ?

We have used commercial  $H_2O_2$  which is stabilized. However, we get good agreement between the hydrocarbon decay method, when other OH sources are used, e.g. (CH3)3COOH (Winiberg PhD thesis, University of Leeds 2014), which is supplied without a stabilizer.

p10, I10-12 and Fig 2 : description of the different conditions not clear; Fig2 (c) it says: the temperature is plotted when gas is either sampled from temperature-controlled air from the calibration flowtube or sampling from the HIRAC chamber. However, there is only one symbol for each OH and HO2. Slopes should be provided.

The temperature gradients when heated/cooled air either from the calibration flowtube or HIRAC was sampled were essentially the same. The caption has been altered to explain this and to provide the slopes of the various plots.

**Figure 2:** (a) Schematic of FAGE Cell showing locations of thermocouples. (b) Internal cell temperatures ( $T_{OH}$  or  $T_{HO2}$ ) and inlet temperatures ( $T_{inlet}$ ) plotted as a function of the external temperature ( $T_{ext}$ ), when sampling air at 293 K from the calibration flowtube. Slope  $T_{inlet} = 0.558 \pm 0.010$ ; Slope  $T_{OH} = 0.497 \pm 0.008$ ; Slope  $T_{HO2} = 0.236 \pm 0.033$ . (c) Internal temperatures as a function of the external temperature when sampling temperature controlled air from the calibration flowtube. Slope  $T_{OH} = 0.890 \pm 0.004$ ; Slope  $T_{HO2} = 0.316 \pm 0.007$  (sampling from the HIRAC chamber gave lines with essentially the same gradients).

p12, I26 : sum of non-OH first order processes: provide examples 'E.g. heterogeneous loss and dilution' has been added.

Fig 3a : could you add measurement uncertainties ? Revised Figure and caption to include examples of errors for each method.

**Figure 3:** a) Time series of [OH] derived from FAGE measurements and from mass spectrometric measurements of cyclohexane removal recorded following  $H_2O_2$  photolysis at 293 K and 1000 mbar air. The error bars shown represent absolute uncertainties in the calibration methods, see Table 4.

p14, I10: could you comment the results obtained without mixing fans? Material added on page 15, lines 5-7.

'Without the fans, the value of  $k_{\text{loss}}$  was reduced, but agreement between the HO2 calibration methods was comparable (within 10%). As HIRAC is generally operated with fans on, we have only reported these data.'

p14, I27: What is the order of magnitude of the percentage of HO2 typically lost to the walls rather than through self-reaction? With the fans on, the first order loss rates where  $0.032 - 0.073 \text{ s}^{-1}$ . The contribution to the mixed HO2 decay will depend on the fraction of HO2 removed. As long times it will be ~90%, however that the beginning of the decay it will be approximately 15 - 50% ( $k_{\text{HO2+HO2}} \times [\text{HO}_2]_0$ ; typically [HO2]0 = 4 x 1010 - 1 x 1011, so  $k' = 0.114 - 0.285 \text{ s}^{-1}$  Following text has been added: (Page 15 lines 7&8)

Wall loss typically contributes 10 - 50% of the initial decay but is well defined in the fitting procedure.

p16, l2: be Corrected

p16, I4: relative to the value at 293 K (to add) Added

p17, figure 5: I cannot see that  $C_{HO2,obs}$  data are more scattered. It looks like there is an increase at the beginning and then a plateau. What is the red line in the HO2 plot? Useful ? Wouldn't it be interesting to show uncertainty only due to temperature variation in addition to the total uncertainty? At some point the graph file became corrupted and we have plotted the same data, but still have the original fit. Not sure how this happened and apologies. The corrected graph has been inserted and is shown below.

p18, l22 : data points. Why only 3? Why are there more OH points than HO2 points? There is a comment in the footnote of Table 2 regarding the reason why there are only 3 data points for  $HO_2$  – malfunction of flowcontroller.

p20: is the concentration of CH2O low enough to avoid HO2 loss through the equilibrium reaction HO2 + CH2O? In the absence of the fans, the HO2 decays are strictly  $2^{nd}$  order demonstrating the absence of any significant loss via HCHO. This could be an issue at lower temperatures limiting the temperature range over which this calibration method would work.

p20, I1:importance Confirmation that chemical removal of the hydrocarbon is dominated by OH reaction is the primary assumption of the alternative OH calibration method. We can account for wall loss and dilution but we have to assume that once the lamps are switched on, chemical removal is by reaction with OH. In such a simple system and with saturated alkanes, one would expect removal to be via OH – still it is always good to have this confirmed! We have added (Page 21 line 2):

A key assumption of the hydrocarbon decay calibration method is that the OH is chemically removed by OH.

Table S5 : explain k'rel Footnote added to Table S5, k' rel =  $k_q$  rel × [Q] rel

**Reviewer 2**

This paper presents the results of measurements of the temperature dependence of different techniques for the calibration of a laser-induced fluorescence – fluorescence assay by gas expansion (LIF-FAGE) instrument for measurements of OH and HO2. The authors compare the calibration factors obtained by the traditional water-vapor photolysis technique as a function of temperature with that obtained by hydrocarbon decay by reaction with OH for OH calibration, and measurement of HO2 decays due to self-reaction for HO2 calibration. The authors find that all calibration methods had a positive temperature dependence. After accounting for known temperature dependent factors, such as number density, quenching and rotational population of the probed level, the authors conclude that the remaining temperature dependence is most likely due to a decrease in the wall loss of radicals in their instrument as the temperature increases, improving the transmission of radicals from the pinhole inlet to the detection axis.

As noted by the authors, these results depends on the design and materials of the FAGE instrument and may not be applicable to other FAGE instruments. However, the results are of interest to the HOx measurement community and suitable for publication in AMT after the authors have considered the following minor comments.

**Once again, we are grateful for the positive comments of the reviewer.**

 As mentioned in the manuscript, the calibration factor will have a dependence on water vapor due to quenching of the fluorescence. The authors illustrate the OH calibration factor at a constant concentration of water, but there is no mention of how the authors account for the water vapor dependence of the calibration factor. The authors state that the HO2 experiments were done in dry conditions to minimize enhancement of the HO2 self-reaction by water vapor. I assume that the water vapor calibration factor was corrected to account for quenching by water vapor when compared to the calibration factors derived by the other techniques, but the authors should clarify how this was done. Did the authors measure the calibration factor as a function of water vapor in order to correct the calibration for dry conditions?

The reviewer is correct here. We have added some additional material towards the end of sections 3.2 (OH) and 3.3 (HO2) explaining the minor corrections made.

Section 3.2. (Page 13, lines 9 – 12)

The HIRAC FAGE system shows a slight sensitivity to water vapour concentrations due to quenching (Winiberg, 2014). Minor corrections (<5%) were made to account for the different water vapour concentrations in the two calibration methods.

Section 3.3 (Page 15 lines 9 - 11)

As with OH detection, minor corrections have been made for the slightly different sensitivities of the system under the different water concentrations of the two calibration methods (Winiberg, 2014).

- There appears to be an error in Figure 5b, as the points do not reflect the values in Table 2, although the fit does. As with reviewer 1, we apologise that somehow the same graph has been included for both Figure 5a and 5b.
- The authors should clarify how the fits of the temperature dependence shown in Figures 5-7 were obtained. Are these bivariate fits weighted by the measurement uncertainty? Captions to Figures 5 7 have been altered to state that lines are the weighted fits to the data points.
- The errors in Table 4 should be clarified are these 1 sigma uncertainties?
  Additional footnote added to Table 4: a Where the error is statistical, it is reported at the 1σ level.